

# Validation of COI metabarcoding primers for terrestrial arthropods

Vasco Elbrecht[1], Thomas W.A. Braukmann[1], Natalia V. Ivanova[1,2],
Sean W.J. Prosser[1], Mehrdad Hajibabaei[1,2], Michael Wright[1,2],
Evgeny V. Zakharov[1,2], Paul D.N. Hebert[1,2] and Dirk Steinke[1,2]

[1] Centre for Biodiversity Genomics, University of Guelph, Guelph, ON, Canada
[2] Department of Integrative Biology, University of Guelph, Guelph, ON, Canada

## ABSTRACT

Metabarcoding can rapidly determine the species composition of bulk samples and thus aids biodiversity and ecosystem assessment. However, it is essential to use primer sets that minimize amplification bias among taxa to maximize species recovery. Despite this fact, the performance of primer sets employed for metabarcoding terrestrial arthropods has not been sufficiently evaluated. This study tests the performance of 36 primer sets on a mock community containing 374 insect species. Amplification success was assessed with gradient PCRs and the 21 most promising primer sets selected for metabarcoding. These 21 primer sets were also tested by metabarcoding a Malaise trap sample. We identified eight primer sets, mainly those including inosine and/or high degeneracy, that recovered more than 95% of the species in the mock community. Results from the Malaise trap sample were congruent with the mock community, but primer sets generating short amplicons produced potential false positives. Taxon recovery from both mock community and Malaise trap sample metabarcoding were used to select four primer sets for additional evaluation at different annealing temperatures (40–60 °C) using the mock community. The effect of temperature varied by primer pair but overall it only had a minor effect on taxon recovery. This study reveals the weak performance of some primer sets employed in past studies. It also demonstrates that certain primer sets can recover most taxa in a diverse species assemblage. Thus, based our experimental set up, there is no need to employ several primer sets targeting the same gene region. We identify several suitable primer sets for arthropod metabarcoding, and specifically recommend BF3 + BR2, as it is not affected by primer slippage and provides maximal taxonomic resolution. The fwhF2 + fwhR2n primer set amplifies a shorter fragment and is therefore ideal when targeting degraded DNA (e.g., from gut contents).

Corresponding author
Vasco Elbrecht,
luckylion07@googlemail.com

## INTRODUCTION

Over the past decade, two methodological and technological advances have made it possible to address the urgent need for the capacity to undertake large-scale surveys of biodiversity (*Vörösmarty et al., 2010*; *Dirzo et al., 2014*; *Steffen et al., 2015*). First, the emergence of DNA barcoding which uses sequence variation in short, standardized gene regions (i.e., DNA barcodes) to discriminate species, has made it possible to quickly and reliably characterize

species diversity (*Hebert et al., 2003*). Second, high-throughput sequencers (HTS) permit the inexpensive acquisition of millions of sequence records (*Reuter, Spacek & Snyder, 2015*). The coupling of HTS with DNA barcoding, commonly known as metabarcoding, allows for characterization of biodiversity at unprecedented scales (*Creer et al., 2016*) as shown by studies of terrestrial (*Gibson et al., 2014*; *Beng et al., 2016*), freshwater (*Hajibabaei et al., 2011*; *Carew et al., 2013*; *Andújar et al., 2017*), and marine (*Leray & Knowlton, 2015*) ecosystems.

Metabarcoding studies on bulk collections of animals usually target a subset of the 658 bp cytochrome *c* oxidase subunit I (COI) ''Folmer'' region (*Folmer et al., 1994*; *Andújar et al., 2018*). This gene region has gained broad adoption because of a rapidly expanding reference database (*Ratnasingham & Hebert, 2007*; *Porter & Hajibabaei, 2018b*) and its good taxonomic resolution (*Meusnier et al., 2008*). Ribosomal markers have been suggested as an alternative (*Deagle et al., 2014*; *Marquina, Andersson & Ronquist, 2018*) because their slower rate of evolution results in more conserved motifs/regions aiding the design of universal primer sets. However, arthropod reference databases for ribosomal markers are very limited for most taxonomic groups (*Clarke et al., 2014*) and ribosomal primer sets often show no substantial improvement in taxon recovery over well-designed COI primer sets (*Elbrecht et al., 2016*; *Clarke et al., 2017*; *Elbrecht & Leese, 2017*; *Krehenwinkel et al., 2017*).

An important consideration for metabarcoding studies is the primer combination used for amplification of the target fragment. It is critical that primer sets optimally match the template sequences of the target species. Mismatches between primer and template can skew read abundance and lead to a substantial bias in taxon detection (*Piñol et al., 2014*; *Elbrecht & Leese, 2015*). Failure to minimize amplification bias reduces the amount of taxa detected in a sample (*Elbrecht & Leese, 2017*). Furthermore, insufficient sequencing depth and/or low DNA concentration can introduce stochastic effects that additionally bias taxon recovery (*Barnes & Turner, 2015*; *Leray & Knowlton, 2017*).

The effectiveness of primer sets can be evaluated by *in vitro* tests with mock communities (*Elbrecht & Leese, 2015*; *Brandon-Mong et al., 2015*; *Leray & Knowlton, 2017*) or by *in silico* tests (*Clarke et al., 2014*; *Elbrecht & Leese, 2016*; *Piñol, Senar & Symondson, 2018*; *Bylemans et al., 2018b*; *Marquina, Andersson & Ronquist, 2018*). The failure to evaluate primers can seriously compromise data quality. For instance, a primer set (*Zeale et al., 2011*) often employed for analyzing gut contents of insect predators (see references in *Jusino et al., 2018*) lacks degeneracy, leading to poor taxon recovery (*Brandon-Mong et al., 2015*). The use of multiple primer sets or even multiple marker genes was proposed to improve taxon recovery (*Alberdi et al., 2017*; *Zhang et al., 2018*). This approach may be optimal for samples of very phylogenetically divergent groups such as protists (*Pawlowski et al., 2017*) or marine benthic communities (*Cowart et al., 2015*; *Wangensteen et al., 2018*; *Drummond, 2018*). However, given the increased cost and time associated with amplifying and sequencing additional markers (*Bohmann et al., 2018*; *Zhang et al., 2018*), the use of multiple primer sets is unnecessary for taxonomic groups with limited diversity. Additionally, relative read abundance comparisons between primer sets will become difficult, as each is subjected to different primer biases (*Elbrecht & Leese, 2015*). We hypothesize that in the case of

## Experimental outline — A

- 36 Candidate primer sets (Fig. 2, primer sequences Tab. S2)
- Two insect bulk samples for testing:

| 374 taxa mock insect community (from Braukmann *et al.* 2019) | One complete malaise trap, 1 week July, Ontario, Canada |

## Primer shortlist — B

- Gradient PCR, annealing 46 - 64.5 °C (Fig. S7 & S8)
- Discarding primer sets with poor amplification or no plateau in PCR gradient

## Metabarcoding (46°C annealing) — C

- Fusion primer design for 21 primer sets
- Two step PCR at 46° annealing temperature
- Two Illumina MiSeq sequencing runs (v3 300 bp Paired End):

| Run I: "mock community" | Run II: "malaise trap sample" |

- Objective: Assess and compare performance of different primer sets (Fig. 3 & 4)

## Mock sample gradient metabarcoding — D

- Fusion primer design for 4 representative primer sets
- Two step PCR at different annealing temperatures: 40.0, 41.6, 43.7, 46.0, 48.5, 50.8, 53.0, 54.7 and 56.0 °C
- MiSeq run III: "gradient"
- Objective: Optimise annealing temperature for the best performing primer sets (Fig. 5)

## Primer recommendations — E

**Figure 1  Overview of the experimental design.** The performance of 36 primer pairs was tested via gradient PCRs with a mock community of insects (A). The 21 pairs that showed best amplification results (B) were selected for further DNA metabarcoding runs utilizing both the mock community and a Malaise trap sample (C). Based on the metabarcoding results, four primer sets showing the good performance were selected for a third test that examined the effects of varying annealing temperatures on taxon recovery and non-specific amplification (D). Based on all results, the optimal primer sets were designated (E).

terrestrial arthropods a single well-designed primer set can be sufficiently effective, and the use of multiple primer sets is not necessary.

This study compares the performance of commonly used and newly developed primer sets on the recovery of species in a bulk DNA extract from 374 insect species (*Braukmann et al., 2019*) and from a Malaise trap sample. Based on a hierarchical testing scheme (Fig. 1)

using gradient PCRs and assessing species recovery with metabarcoding, we selected four primer pairs whose metabarcoding performance was tested on a range of annealing temperatures.

## MATERIAL AND METHODS

### Tested samples and experimental outline

We used two samples to test a range of primer sets for metabarcoding: a mock community of 374 species (*Braukmann et al., 2019*) and a sample collected with a Malaise trap (Fig. 1). The mock community is comprised of 374 species (Fig. 1A), and each specimen represented by an individual BIN (taxonomic breakdown shown in Fig. S1A, *Ratnasingham & Hebert, 2013*). A detailed list of specimens and their Barcode of Life Datasystems process IDs (BOLD, *Ratnasingham & Hebert, 2007*) is given in Table S1. For most specimens, the full 658 bp barcode region was available through BOLD, but we completed reads for three taxa with shorter sequences by extracting haplotypes from our metabarcoding data using a denoising approach (*Elbrecht et al., 2018b*). The resulting reference library is available as a fasta file (See Scripts S1). To compare mock community results with a field sample, we collected insects with a Townes-style Malaise trap (Bugdorm, Taiwan) deployed in a grassland/forest area near Waterloo, Ontario, Canada (43°29′30.8″N80°36′59.6″W). We selected a single weekly sample (June 30–July 7, 2018) and dried it for three days in a disposable grinding chamber. The sample was ground to fine powder using an IKA Tube Mill control (IKA, Breisgau, Germany) at 25,000 rpm for 2 × 3 min. DNA was extracted from 21 mg of ground tissue using the DNeasy Blood & Tissue kit (Qiagen, Venlo, Netherlands).

The mock community DNA extract was used to test 36 primer pairs by comparing amplification success across a range of annealing temperatures. Twenty-one primer pairs whose amplicon concentrations plateaued in amplicon concentration at lower annealing temperatures were selected for metabarcoding both the mock community and the Malaise trap sample. Four representative primer sets showing high success in species recovery were selected to determine the optimal annealing temperature for maximizing species recovery from bulk samples (see Fig. 1).

### Gradient PCRs

Thirty-six primer combinations commonly used for metabarcoding were selected for gradient PCR (Fig. 2), to assess primer efficiency at different annealing temperatures. Some of these primers represent new combinations, as well as new variants of primers by shifting the primer binding site by 3 bp, by incorporating degeneracy, or by replacing inosine "I" with "N" and vice versa. PrimerMiner v0.18 was used to generate an alignment visualization (*Elbrecht & Leese, 2016*) using reference sequences for 31 arthropod orders downloaded and aligned as part of an earlier study (*Vamos, Elbrecht & Leese, 2017*). The plot of the full alignment with binding sites for all primers used in this study is available in Fig. S2.

Mock community gradient PCRs for 36 primer combinations were run on an Eppendorf Mastercycler pro (Hamburg, Germany). PCRs were set up with 2× Multiplex PCR Master
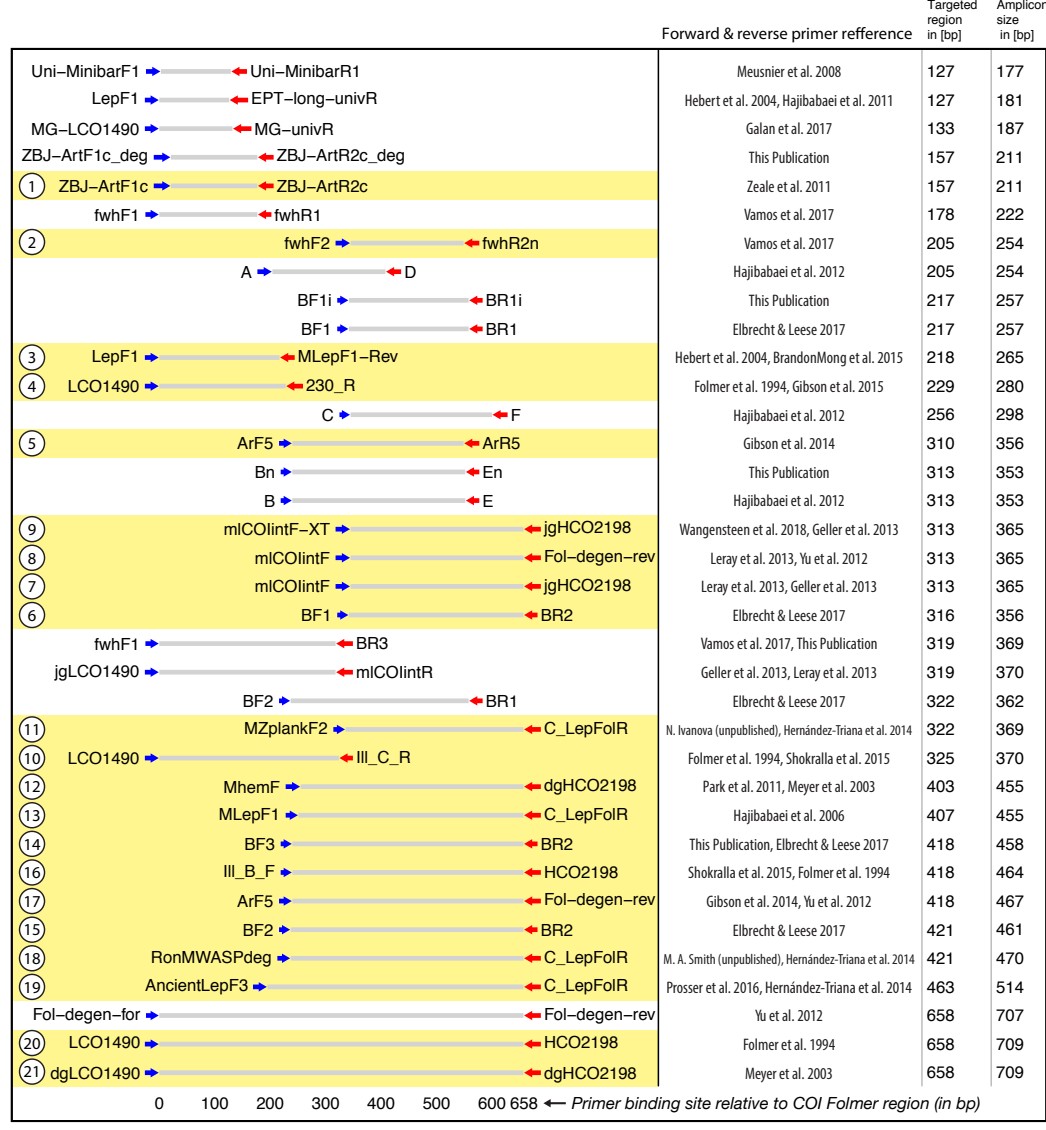

**Figure 2** **Target and amplicon length for the 36 primer sets evaluated via gradient PCR.** The 21 primer sets selected for sequencing are highlighted in yellow while an ID for each pair is shown on the left. See Table S2 for primer sequences. Primer references: *Folmer et al., 1994*; *Meyer, 2003*; *Hebert et al., 2004*; *Hajibabaei et al., 2006*; *Meusnier et al., 2008*; *Hajibabaei et al., 2011*; *Zeale et al., 2011*; *Park et al., 2011*; *Yu et al., 2012*; *Hajibabaei et al., 2012*; *Leray et al., 2013*; *Geller et al., 2013*; *Gibson et al., 2014*; *Hernández-Triana et al., 2014*; *Shokralla et al., 2015*; *Brandon-Mong et al., 2015*; *Gibson et al., 2015*; *Prosser et al., 2015*; *Elbrecht & Leese, 2017*; *Vamos, Elbrecht & Leese, 2017*; *Wangensteen et al., 2018*; *Galan et al., 2018*.

Mix Plus (Qiagen, Hilden, Germany), 0.5 µM of each primer (IDT, Skokie, Illinois), 12.5 ng DNA, and molecular grade water (HyPure, GE, Utha, USA) for a total volume of 25 µL. One BF2 + BR2 primer (*Elbrecht & Leese, 2017*) positive control using the mock community DNA and one negative control were included with each primer set.

The following thermocycling protocol was used: initial denaturation at 95 °C for 5 min then 29 cycles of denaturation at 95 °C for 30 s followed by a gradient of annealing

temperatures from 44.5 –64.5 °C for 30 s with extension at 72 °C for 50 s, and a final extensions of 5 min at 72 °C. PCR success and fragment length were determined by visualizing amplicons on a 1% agarose gel. Amplicon concentration was quantified without prior cleanup using a High Sensitivity dsDNA Kit on a Qubit fluorometer (Thermo Fisher Scientific, MA, USA).

## Primer selection for metabarcoding

Based on the results of gradient PCR (Fig. 1B), we selected 21 primer sets for metabarcoding that showed strong, consistent amplification and reached plateau in amplicon concentration at lower annealing temperatures (minimum of 0.6x amplification at 50.2 °C, Fig. S8). A few primer sets generated amplicons at 46 °C, but were excluded because they failed to reach an asymptote in concentration at lower annealing temperatures.

## Metabarcoding (mock community and malaise trap)

21 primer sets for both the mock community and the Malaise trap sample were selected for DNA metabarcoding and Illumina MiSeq sequencing. We employed a fusion primer based two-step PCR protocol that amplifies target fragments in the first step and attaches in-line tags and Illumina TruSeq library sequence tails during the second PCR (*Elbrecht & Steinke, 2018*). To ensure sufficient sequence diversity for sequencing we used in-line tags of different length to shift amplicons against each other and sequenced half of the samples in reverse orientation by swapping the Illumina P5 and P7 sequencing adapters on the fusion primers (*Elbrecht & Leese, 2015*). The 7 bp tags with different insert lengths were randomly generated using R scripts (*Elbrecht & Steinke, 2018*), but were subsequently manually edited to maximize the Levenshtein distance between tags (Fig. S3). Figure S4 shows the fusion primer sequences used for library preparation.

For the first PCR step, we used the same protocol as for the gradient PCR but used a fixed annealing temperature of 46 °C and 24 cycles of amplification. One negative control containing the BF2 + BR2 primer combination and one containing no primers were included in the PCR (see Table S2 for primer list).

For the second PCR step, one μL of the PCR product generated by each primer set was used as template (with no quantification or reaction cleanup) under similar PCR conditions to the first PCR step except the extension time was increased to 2 min while the number of cycles was reduced to 14.

PCR products were cleaned using SPRIselect (Beckman Coulter, CA, USA) with a sample to volume ratio of 0.76×. DNA concentration was quantified using a Qubit fluorometer, High Sensitivity dsDNA Kit (Thermo Fisher Scientific, MA, USA). Subsequently, individual libraries were equimolar pooled following adjustment for amplicon length (Table S1).

The mock community library was sequenced on an Illumina MiSeq with 300 bp paired end sequencing (v3 chemistry) with a 5% PhiX spike in. Amplicons for the Malaise sample were generated with half the DNA amount (6.25 ng) to reduce the chance of PCR inhibition and 29 cycles for the first PCR step. Individual libraries were pooled equimolar, but we factored in the preference for shorter reads by Illumina sequencing using the mock community sequencing results (Fig. S9, Table S1). The Malaise sample was also sequenced

on an Illumina MiSeq with 300 bp paired end sequencing (v3 chemistry) with a 5% PhiX spike in.

For both sequencing runs, PCR negative controls where omitted, as additional fusion primers would have been needed for sample tagging. For the same reason, we also decided against PCR replicates, given that DNA metabarcoding results are usually highly reproducible (*Elbrecht et al., 2017*; *Braukmann et al., 2019*) and potential tag switching could be easily identified based on the different primer sets used (i.e., amplicon length).

## Bioinformatic processing

Quality control of raw sequence data was done with FastQC v0.11.7 and multiQC v1.4 (*Ewels et al., 2016*). Sequence data was first demultiplexed and processed with the R wrapper script JAMP v0.68 (https://github.com/VascoElbrecht/JAMP). Reads were paired-end merged using Usearch v11.0.667 (*Edgar, 2010*), allowing for 99 mismatches or 75% similarity between overlapping regions to maximize the amount of merged reads. Primer sequences were subsequently trimmed using cutadapt v1.18 with default settings (*Martin, 2011*). Reads deviating by more than 10 bp from the expected amplicon length were discarded. Usearch (*Edgar & Flyvbjerg, 2015*) was used to remove reads with an expected error probability of 1 or higher, and to dereplicate and map reads against the 374 reference sequences of the mock community (usearch_global with minimum 97% identity). Resulting tables were automatically summarized into a hit table of all samples using the function map2ref implemented in JAMP. The hit table was subsampled using a custom R script (Scripts S1) to determine the number of taxa detected at different sequencing depths. Figure 3 provides an overview of the results of the processing steps and all scripts are available in Scripts S1.

Data for the Malaise sample was processed using the same pipeline but mapped against a reference database consisting of public sequence records for arthropods found in Ontario (downloaded from BOLD December 2018). Gaps and terminal Ns were removed from all sequences. Sequences outside the length range of 648–668 bp were discarded (Scripts S1). Reads were mapped against this reference database using map2ref, but singletons in each sample were discarded and mapping required a 99% match and maxaccepts=0, maxrejects=0, to reduce the number of false positives. Reads matching to the same Barcode Index Number (BIN, *Ratnasingham & Hebert, 2013*) were collapsed and reads that matched reference sequences that lacked a BIN assignment were merged based on taxonomy, and combined into 11 MOTUs.

## Gradient metabarcoding

Out of the 21 metabarcoded primer sets, we selected four representative primer sets of different amplicon length that recovered over 95% the mock community (ArF5 + Fol-degen-rev, BF3 + BR2, mlCOIintF + Fol-degen-rev and fwhF2 + fwhR2n, Fig. 1D) to evaluate the impact of nine annealing temperatures (40.0, 41.6, 43.7, 46.0, 48.5, 50.8, 53.0, 54.7 and 56.0 °C) on taxon recovery. Temperatures below 46 °C were specifically chosen to explore the impact of non-specific amplification. Other than running the first and second PCR step as gradient PCRs, all laboratory conditions and bioinformatic steps were identical to the prior mock community metabarcoding run. For tagging samples in
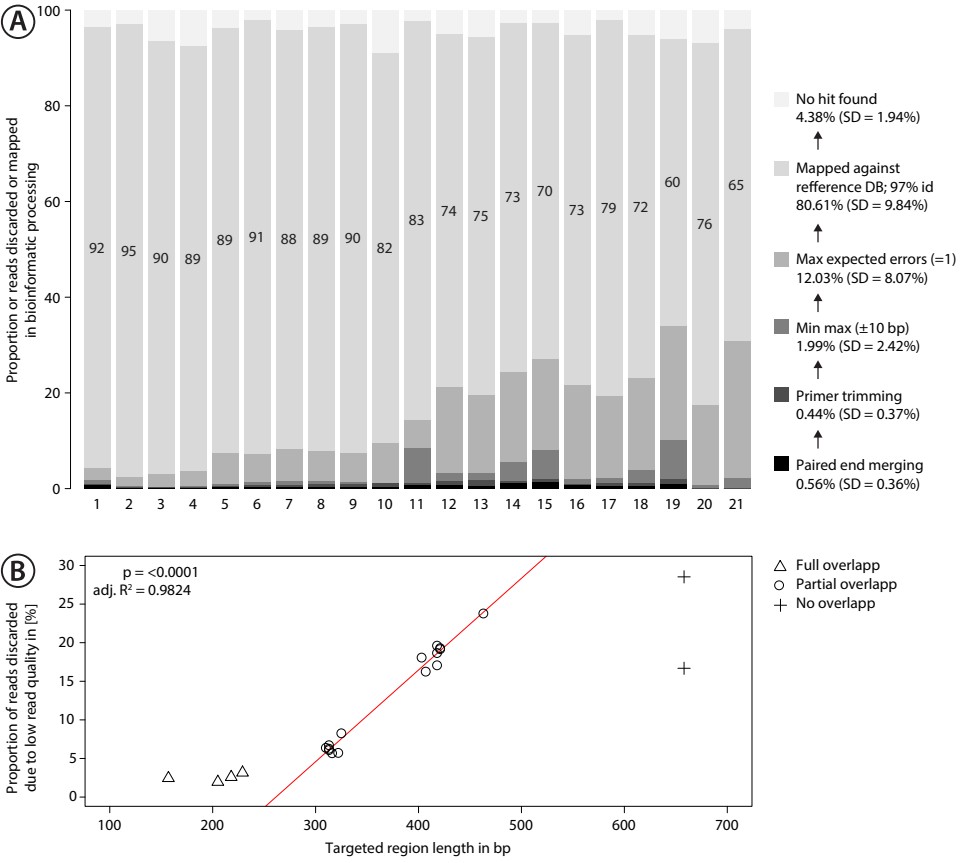

**Figure 3** **Proportion of sequences discarded or mapped to reference sequences in the mock community.** (A) Bar plots show the relative proportion of reads that were discarded or mapped. Numbers in bars indicate the proportion of reads that matched one of the 374 species in the mock community. The number for each primer pair on the *x*-axis corresponds with that in Fig. 2. (B) Proportion of sequences discarded by max expected errors = 1 filtering using Usearch, plotted against the length of the target region (in bp). Red line indicates linear regression.

the second PCR step, additional fusion primers were developed (Fig. S5) and checked for sufficient Levenshtein distance (Fig. S6, *Elbrecht & Steinke, 2018*). Individual samples were equimolar pooled, and the library sequenced using an Illumina MiSeq with 300 bp paired end sequencing (v3 chemistry) and a 5% PhiX spike in. Bioinformatic analysis was identical to the previous mock community MiSeq run at 46 °C annealing temperature.

## Statistical analysis

For statistical analysis R v3.5.0 was used and all scripts used to generate figures are available in Scripts S1. To simulate the effect? of equal sequencing depth at 10,000 and 100,000 reads with each primer set, hit tables were subsampled 1,000 times and number of BINs or taxa recovered was averaged. The relative abundance of reads per taxon (above 0.001%) for each of the 21 primer sets (Table S1) tested with the mock community was analysed using a Principal Component Analysis implemented in the R package FactoMineR v1.34. The same data was used to visualize the similarity between communities recovered with each

primer set, using the R package vegan v2.5-2. A dendrogram was generated using both Jaccard similarity and Bray–Curtis dissimilarity.

## RESULTS

### Gradient PCR results and primer set selection

All primer sets generated amplicons with the expected length (Fig. S7) although a few amplicons showed faint secondary bands following the gradient PCR. None of the negative controls showed visible bands. Amplicon concentrations (ng/µl) reached an asymptote for 21 of the 36 primer sets (58%) at <50 °C and they were selected for sequencing (Fig. S8). While some other primers showed clear bands in the agarose gel (Fig. S7), they were excluded from sequencing because of their limited annealing temperature range.

Amplification success for newly designed primers showed mixed results (Figs. S7 and S8). A more degenerate version of ZBJ − ArtF1c + ZBJ − ArtR2c decreased amplification efficiency. Substituting N for inosine led to increased amplification efficiency for BF1i + BR1i, while replacing inosine with N reduced amplification efficiency for Bn + En. The binding site of the BF2 primer was shifted 3 bp forward (BF3) to reduce slippage effects (*Elbrecht, Hebert & Steinke, 2018a*). In combination with the BR2 primer set, both versions showed similar amplification efficiency.

### Metabarcoding and bioinformatic processing

Sequencing of the mock community tested with 21 primer sets on MiSeq (300 bp PE) produced 24,348,000 reads of good quality (Q30 $\leq$ 85.8% of reads). The PCR negative control (not included in sequencing) showed no visible band on a agarose gel. Raw sequence data is available on NCBI SRA via accession number SRX4908948. Sequencing depth was negatively correlated with amplicon length (Fig. S9 , linear regression, $p < 0.0001$), with at least 280 K sequences per sample. The number of discarded sequences after data processing varied among primer sets (Fig. 3A); on average 80.61% (SD = 9.84%) of the reads were mapped to the 374 reference sequences. For the primer sets MZplankF2 + C_LepFolR, BF3 + BR2, BF2 + BR2 and AncientLepF3 + C_LepFolR more than 3% of the amplicons deviated by more than 10 bp from the expected amplicon length (Fig. 3, Fig. S10). Primer combinations involving mlCOIintF, BF1, BF2 and fwhF2 showed length variation of 1-2 bp base pairs (Fig. S10). Additionally, an average of 12.03% (SD = 8.07%) of all reads were discarded through expected error quality filtering (max ee = 1, Fig. 3A). In particular, longer amplicons with little or no overlap in paired end sequencing were affected (Fig. 3B). Quality filtered read data mapped against reference sequences is provided in Table S1.

The Malaise sample yielded 16,629,020 reads of good quality (Q30 $\leq$ 92.59% of reads). Raw sequence data is available on NCBI SRA via accession number SRX5175597. The PCR negative control was not included in sequencing, but showed no visible band on a agarose gel. Sequencing depth was positively correlated with amplicon length (linear regression, $p = 0.0004$, Fig. S9), but there was a reduced length bias in comparison to the mock community sequencing run (Fig. S9).

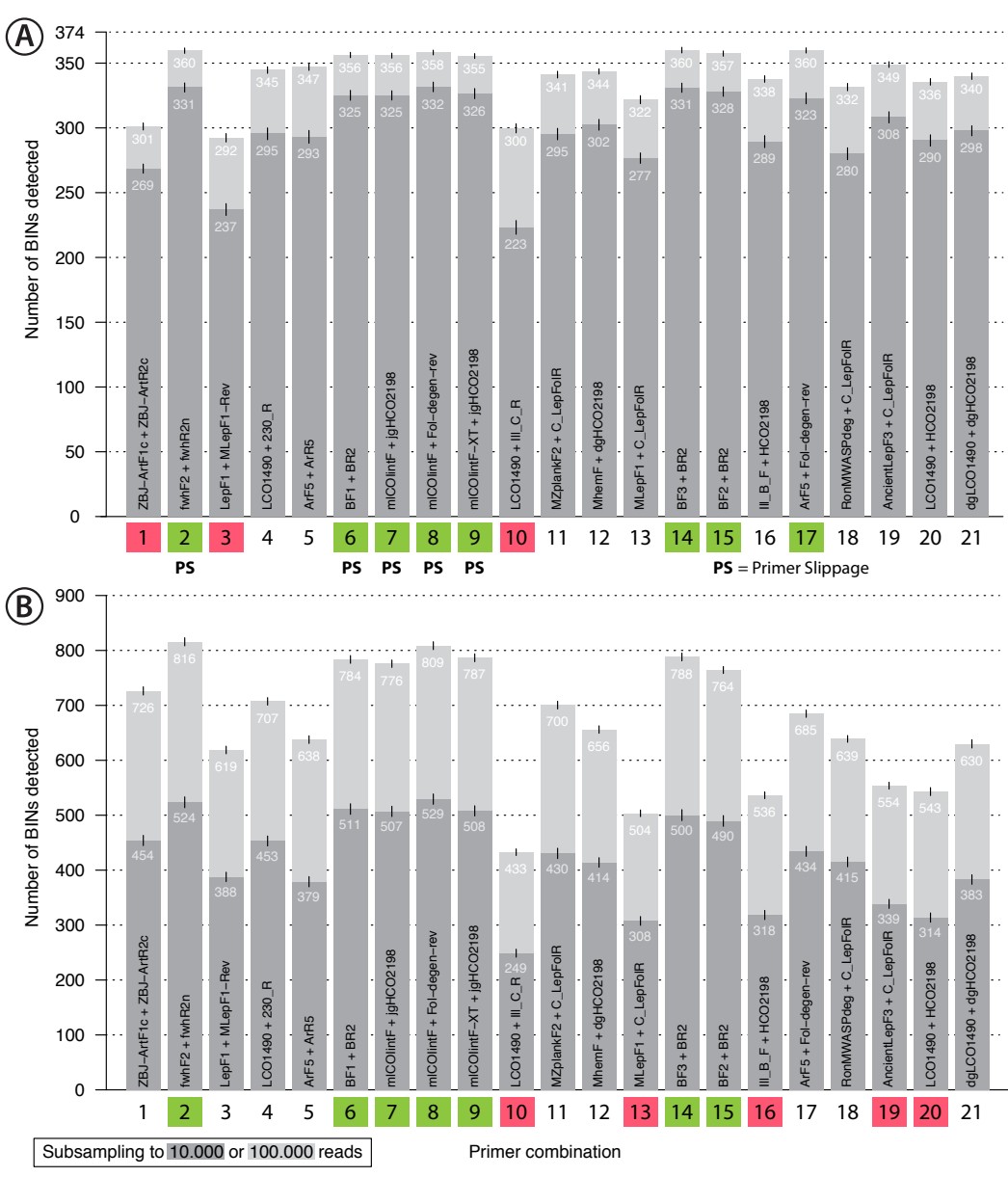

**Figure 4** **Bar plot showing the number of BINs recovered using metabarcoding with 21 primer pairs.** The dark grey bar indicates subsampling at 10,000 reads while the light grey bar indicates subsampling at 100,000 reads per sample, each run with 1,000 replicates. Error bars show the standard deviation. Primer combinations affected by primer slippage (*Elbrecht, Hebert & Steinke, 2018a*) are marked with "PS". (A) Mock sample data, with primer combinations highlighted in green that detected more than 350 of the 374 BINs, while those that recovered fewer than 310 BINS are highlighted in red. (B) Malaise trap data - primer combinations highlighted in green detected more than 750 BINs while those highlighted in red detected less than 600 BINs.

## Primer performance and BIN/species recovery with metabarcoding

Recovery of the mock community was high for most primer sets with an average of 91% of the 374 species recovered (SD = 0.64%, subsampling to 100,000 reads, Fig. 4A). With decreasing sequencing depth, recovery diminished, as shown by rarefaction curves (Fig. S11). The primer sets ZBJ-ArtF1c + ZBJ-ArtR2c, LepF1 + MLepF1-Rev and LCO1490 + Ill_C_R showed poor species recovery in comparison with the other primers. Interestingly, rarefaction analysis showed no strict relationship between recovery and primer degeneracy. For example, LCO1490 + HCO2198 had no degenerate sites but had good recovery (90% of taxa). However, primers that lacked degeneracy often had low amplification success and detected fewer species than primer sets with degeneracy (Fig. S12). The primer combinations fwhF2 + fwhR2n, BF1/BF2/BF3 + BR2, ArF5 + Fol-degen-rev and those based on mlCOIintF and its derivatives showed the best performance with similar recovery rates (recovery ≥ 95% of the community, Fig. 4, Fig. S12). Taxa recovery was consistent across orders, except for Hymenoptera which were often recovered with lower read counts (Fig. S1A). A Principal Component Analysis (PCA) of relative taxon recovery shows that primer combinations with similar taxon recovery tended to cluster together (Fig. S13), although only 29.36% of variability can be explained by both components. Jaccard similarity and Bray-Curtis based dendograms (Fig. S14) illustrate that recovery is generally similar among primers, but that combinations with poor species recovery tend to cluster together.

Sequencing of the Malaise sample confirmed the strong performance of some primer sets, but others showed lower species recovery (Fig. 4B). As the species composition of the Malaise sample was unknown, BIN counts at different sequencing depths were used to estimate taxon recovery for all sets of primers. Heat maps for both the Malaise sample (Fig. S18) and the mock community (Fig. S12) were generally congruent but short amplicons from the Malaise sample detected more taxa present in very low abundance. This trend was also reflected in the number of taxa detected with each primer set (Fig. 4B) because longer amplicons such as Ill_B_F + HCO2198, AcientLepF3 + C_LepFolR or LCO1490 + HCO2198 exhibited lower taxon recovery than shorter fragments. Most primers that performed well for the mock community also did so for the Malaise sample (highlighted in green in Fig. 4B), except the ArF5 + Fol-degen-rev primer set. These patterns were consistent with varying sequencing depths with no asymptote reached in the rarefaction analysis (Fig. S19). Additionally, the rarefaction analysis shows a greater range in the number of taxa detected with different primer sets than for the mock community (Fig. S11). Detection across orders was very consistent for primer sets that show good taxa recovery, while especially Hymenoptera and Hemiptera were underrepresented with primer sets recovering fewer taxa (Fig. S1B).

## Gradient PCR metabarcoding

When the performance of the four good performing primer pairs was analyzed at nine annealing temperatures, 23,770,810 sequences (NCBI SRA; ID: sRX4908947) were obtained with good read quality (Q30 =<82.9% of reads). Sequence coverage averaged 0.58 million (SD = 0.1 million) per sample with a lowest value of 0.38 million reads. The PCR negative

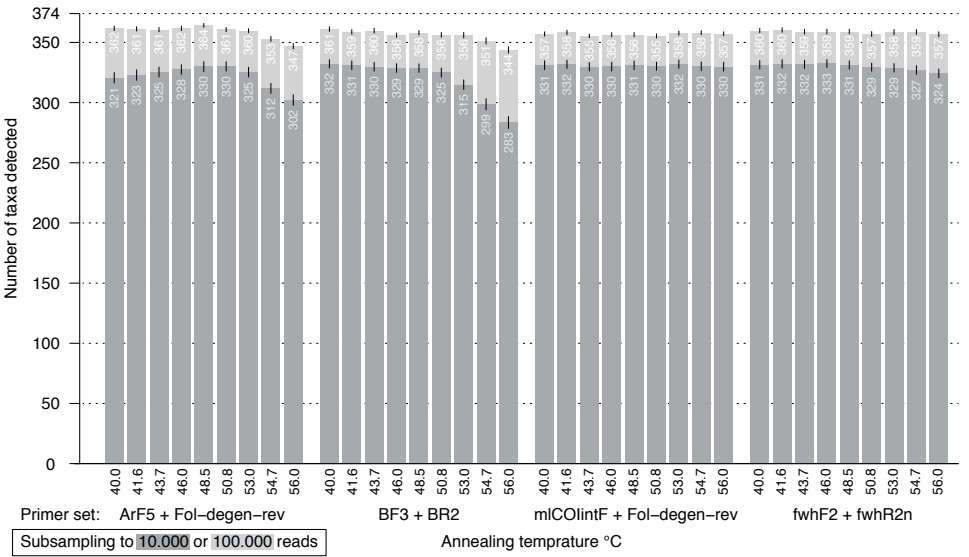

**Figure 5  Bar plot showing the number of BINs recovered from the mock community at different annealing temperatures.** The dark grey bar indicates subsampling at 10,000 reads, while the light grey bar depicts subsampling at 100,000 reads per samples; both were run with 1,000 replicates. Error bars show the standard deviation.

control was not included in sequencing but it showed no visible band on the agarose gel. Results at 46 °C were very similar to the prior metabarcoding run with abundance differences mostly affecting low abundant OTUs (Fig. S15, linear regression adj. $R^2 > 0.97$). Changes in annealing temperature from 40–56 °C only had minor effects on species recovery (Fig. 5). In particular, two primer sets (mlCOIintF + Fol-degen-rev; fwhF2 + fwhR2n) showed little variation in species recovery across the range of annealing temperatures. By comparison, recovery rates decreased at temperatures above ∼53 °C for both BF3 + BR2 and ArF5 + Fol-degen-rev (Fig. 5, Fig. S16). Length variation in amplicons as a result of primer slippage was not temperature dependent, but the BF3 + BR2 primer set generated more short non-target amplicons at lower temperatures (over 1/4 of sequences, Fig. S17).

## DISCUSSION

Using a mock community, we tested a total of 36 different primer combinations, 21 of which were selected for a more detailed metabarcoding analysis. While we did not run replicates for most primer sets, results at 46 °C for gradient metabarcoding and the mock community run were similar. This result is consistent with previous studies on bulk samples which indicated that replicates typically produce similar results (*Elbrecht et al., 2017*; *Braukmann et al., 2019*), particularly when the variation of low abundant OTUs (i.e., <0.001%) introduced by stochastic effects is ignored (*Leray & Knowlton, 2017*). Consequently, for metabarcoding of bulk samples, replication should be done at the sampling level (*Hurlbert, 1984*) rather than using DNA extracts or replicate PCRs. While technical replicates do increase confidence in experimental outcomes and make it easier to

detect cross-contamination (*Zepeda-Mendoza et al., 2016*; *Elbrecht & Steinke, 2018*; *Macher & Weigand, 2018*), they deliver limited information given the substantial increase in cost and laboratory workload. If the detection of rare taxa is important for a project, an increase in sequencing depth (*Smith & Peay, 2014*; *Braukmann et al., 2019*) and use of a tagging system resistant to tag switching (e.g., fusion primers, *Elbrecht et al., 2017*) is a good alternative to replication. Even with the shallow sequencing depth (100,000 reads) used in this study, most primer sets recovered a majority of the taxa in the mock community. This was not necessarily the case for the Malaise trap sample (Fig. S19) which is more diverse than the mock community tested. However, the comparison of taxon recovery at different sequencing depths by the tested primer sets allowed for good benchmarking, without capturing the full community. We were also able to characterize the positive bias of the Illumina MiSeq towards shorter fragments (Fig. S9), which can be offset by adjusted amplicon concentrations when running fragments of different length in the same run (Fig. S9).

## Primer performance

As several primer sets recovered most of the taxa in the mock community in similar proportions (Fig. 4A), our study has identified several suitable primer sets for metabarcoding terrestrial arthropods communities. The exact choice of primer set will depend on the context of a study, required amplicon length and desired taxonomic resolution (*Meusnier et al., 2008*; *Porter & Hajibabaei, 2018a*). For instance, the fwhF2 + fwhR2n primer set produces a 205 bp amplicon that is ideal when targeting degraded DNA in eDNA or gut contents (*Bylemans et al., 2018a*). The BF1 + BR2 and all three mlCOIintF-based primer sets generate slightly longer fragments (316/313 bp), but they are prone to slippage (*Elbrecht, Hebert & Steinke, 2018a*) which can cause problems with sequence denoising

(*Callahan, McMurdie & Holmes, 2017*) during data analysis. We overcame this problem for the longer BF2 + BR2 fragment (421 bp) by moving the BF2 primer 3 bp forward (BF3). The BF3 + BR2 combination as well as the ArF5 + Fol-degen-rev primer set represent good choices for long (>400 bp) COI fragment amplification. The ArF5 + Fol-degen-rev primer set appears to be less affected by non-specific amplification at lower annealing temperatures than the BF3 + BR2 primer pair. Although these longer fragments can improve taxonomic resolution, they show less overlap in Illumina paired end sequencing leading to more reads being excluded during quality filtering (Fig. 3).

We observed an increase of rare taxa detected with short amplicons in the Malaise sample, but these are likely false positives due to the decreased taxonomic resolution of shorter amplicons (*Meusnier et al., 2008*; *Porter & Hajibabaei, 2018a*). Even though the ZBJ-ArtF1c + ZBJ-ArtR2c primer set detected over 700 taxa in the Malaise sample, a value comparable to other well performing primer pairs, it failed to detect abundant BINs that most of the other primer pairs recovered (Fig. S18). Well performing primers showed no bias against specific orders, while less suitable primers did struggle with detection of Hymenoptera and Hemiptera (Fig. S1). The decreased detection of Hymenoptera in the mock community can likely be attributed to the lysis protocol used for DNA extraction

from the insect abdomens (*Braukmann et al., 2019*). This was not the case for the malaise sample, where the bulk sample was ground to a fine powder, making the tissue more accessible to the lysis buffer.

Based on the results of this study we recommend the BF3 + BR2 primer set for metabarcoding of malaise trap samples. The 458 bp long amplicon offers excellent taxonomic resolution and, unlike many of the other tested primer sets, it is not being affected by primer slippage (*Elbrecht, Hebert & Steinke, 2018a*), making it ideal for denoising approaches. Longer amplicons also contain more information about genetic variability and haplotypes when using Exact Sequence Variants (ESVs). While about 20% of the reads generated with this primer set are lost during quality filtering, this can be easily overcome by using more recent Illumina platforms (e.g., NovaSeq) as they provide much higher throughput (*Singer et al., 2019*). However, primer selection might vary with study goal, targeted taxonomic groups, desired taxonomic resolution, DNA quality, and sequencing platform used.

## Primer design

Primer sets with differing degeneracy, inosine inclusion, and differences in the primer binding region showed variable taxonomic recovery making it difficult to establish clear predictors for primer performance. While degeneracy generally improves the universality of a primer (*Krehenwinkel et al., 2017*), some highly degenerate primers performed poorly in our tests (Fig. 4). Additionally, even if a primer set shows good taxon recovery, it can still be susceptible to dimerization, to non-specific amplification, or to primer slippage (*Elbrecht, Hebert & Steinke, 2018a*) (Fig. S10). These complexities are difficult to predict *in silico* and it is important to validate metabarcoding primer sets *in vitro* using taxa and samples from the targeted ecosystems. For example, the BF2 + BR2 primer set generated non-specific amplicons (often bacterial), which can become a serious complication for eDNA studies where target DNA is scarce (*Macher et al., 2018*; *Hajibabaei et al., 2019b.*; *Collins et al., 2019*). High primer degeneracy will likely increase primer universality but decrease specificity. This is less problematic when metabarcoding DNA extracts from bulk specimen samples where target DNA predominates but can be different for environmental DNA samples.

The present study did not reveal if the use of inosine can reduce problems created by high primer degeneracy. Some primers modified with inosine performed well, but others did not. The same was true for highly degenerate primers. However, we did show that for the fusion primer system (*Elbrecht & Steinke, 2018*), primers employed in the second PCR step can be designed with "N" instead of inosine (Fig. S5). This substantially reduces costs when large fusion primer quantities are needed for reliably tagging and sample multiplexing. Primer performance could be further improved by adding degeneracy and/or using inosine, but performance will suffer if too much degeneracy is added. Despite careful primer design following best practices (*Abd-Elsalam, 2003*), primer performance can still vary in its suitability for the primer binding site. A primer that works well on paper, might still not work *in vitro* and we strongly recommend testing primers with a mock community or field sample.

## Annealing temperature

While primer choice is critical for metabarcoding projects, PCR can also be biased by the polymerase used (*Nichols et al., 2018*), cycle number (*Vierna et al., 2017*; *Krehenwinkel et al., 2017*), GC content (*Braukmann et al., 2019*), inhibitors (*Demeke & Jenkins, 2009*; *Sellers et al., 2018*), and annealing temperature (*Aylagas et al., 2016*; *Clarke et al., 2017*; *Krehenwinkel et al., 2018*). It is generally assumed that primers bind better at lower annealing temperatures as potential mismatches between template and primer have less influence. While touchdown PCR does not improve species recovery (*Clarke et al., 2017*), lower annealing temperatures slightly increases recovery (*Aylagas et al., 2016*). Although it seems intuitive that lower annealing temperatures lead to better taxonomic recovery, previous studies explored only a limited temperature range never going below 46 °C, likely due to the increased risk of non-specific amplification. We studied four representative primer pairs at 9 different annealing temperatures across a wider range (gradient PCR from 40–56 °C) and were unable to find a universal effect of annealing temperature. BF3 + BR2, mlCOIintF + Fol-degen-rev and fwhF2 + fwhR2n primer sets are largely unaffected by changes at low annealing temperatures. On the other hand, recovery peaks at 48.5 °C for the ArF5 + Fol-degen-rev primer set. For all four primer pairs, annealing temperatures between 46–50 °C are probably good choices for metabarcoding. However, this highly depends on melting temperature ($T_m$). It is advisable to test newly designed metabarcoding primers across an annealing temperature gradient. However, given that most tested primers did perform similarly well at temperatures usually used for metabarcoding, sequencing gradient PCRs might not always be necessary. Running the four primer pairs at temperatures below 46 °C did not substantially increase taxa recovery, while for some primers it also increased the risk of dimer amplification and occurrence of non-target DNA.

## No need for multiple primer sets

Eight primer combinations (Fig. 4, highlighted in green) each detected 95% or more of the taxa present in the mock community, and most of them could therefore be suitable choices for metabarcoding studies targeting terrestrial arthropods. Of these, seven showed very good performance with the malaise trap sample. This is in stark contrast to earlier studies (*Alberdi et al., 2017*; *Zhang et al., 2018*) recommending the use of multiple primer sets to increase coverage. This discrepancy can be explained by primer choice, because (*Zhang et al., 2018*) used LCO1490 and HCO2198 primers which lack degeneracy, and (*Alberdi et al., 2017*) worked with gut content samples; thus replicates might be substantially affected by stochastic effects resulting from low DNA yield. Additionally, the primers used (ZBJ-ArtF1c + ZBJ-ArtR2c) by *Alberdi et al. (2017)* performed poorly in our study. This particular primer combination (*Zeale et al., 2011*) is widely used for gut content metabarcoding (*Jusino et al., 2018*), but our results show substantial amplification bias, confirming the low taxon recovery observed before for this primer pair (*Brandon-Mong et al., 2015*). An alternative primer pair to analyze gut content from predators consuming insects could be the pair fwhF2 + fwhR2n because it shows better taxonomic recovery.

The use of COI primer sets with limited or no degeneracy such as in (*Zhang et al., 2018*; *Jusino et al., 2018*) is not recommended. In general, careful primer design and validation

(ideally using mock communities) cannot be replaced by the use of multiple COI primer sets (*Alberdi et al., 2017*; *Zhang et al., 2018*) or ribosomal markers (*Deagle et al., 2014*), given the increased workload of a multi marker/primer approach and the limited taxonomic resolution of ribosomal markers (*Clarke et al., 2017*; *Marquina, Andersson & Ronquist, 2018*). These results were also recently confirmed by (*Hajibabaei et al., 2019a*), which showed that the use of multiple primer sets did not substantially improve taxa detection.

## CONCLUSIONS

Our study demonstrates that the fwhF2 + fwhR2n, BF1, BF2, BF3 + BR2 and mlCOIintF based primer sets all perform well when metabarcoding terrestrial arthropod samples. We recommend fwhF2 + fwhR2n for amplification of degraded DNA samples such as gut contents, and BF3 + BR2 when high taxonomic resolution is required. The BF3 + BR2 primer set is also ideal for working with Exact Sequence Variants (ESVs), as it is not affected by primer slippage. For most of these primer sets, annealing temperatures of 46-50 °C are ideal. The present study also reinforces the importance of careful primer validation using mock and field samples, especially when primer performance has not yet been evaluated for the taxonomic group under study. Based on our results, the use of multiple primer sets seems rarely justified as it increases laboratory effort without substantially improving taxon recovery. Our study sets the stage for standardized approaches for large-scale biodiversity analysis. However, given the vast diversity of arthropods in different sampling approaches and geographical regions, primer tests on additional samples are required to fully confirm our findings for large-scale surveys of arthropod diversity.

## ACKNOWLEDGEMENTS

We thank all staff at the CBG who collected the samples employed to assemble the mock community, and Al Woodhouse from the Waterloo District School Board for his aid with Malaise trap sampling. We also thank the Optimist Club of Kitchener-Waterloo for access to their site at Camp Heidelberg. We are grateful to the three reviewers and editor who provided comments that improved our manuscript. This study represents a contribution to the University of Guelph Food From Thought research program.

### Funding
This study was supported by funding through the Canada First Research Excellence Fund. The funders had no role in study design, data collection and analysis, decision to publish, or preparation of the manuscript.

### Grant Disclosures
The following grant information was disclosed by the authors:
Canada First Research Excellence Fund.
## Competing Interests

The authors declare there are no competing interests.

## Author Contributions

- Vasco Elbrecht conceived and designed the experiments, performed the experiments, analyzed the data, contributed reagents/materials/analysis tools, prepared figures and/or tables, authored or reviewed drafts of the paper.
- Thomas W.A. Braukmann performed the experiments, analyzed the data, contributed reagents/materials/analysis tools, authored or reviewed drafts of the paper.
- Natalia V. Ivanova, Sean W.J. Prosser, Mehrdad Hajibabaei, Michael Wright and Evgeny V. Zakharov contributed reagents/materials/analysis tools.
- Paul D.N. Hebert authored or reviewed drafts of the paper.
- Dirk Steinke conceived and designed the experiments, authored or reviewed drafts of the paper.

## DNA Deposition

The following information was supplied regarding the deposition of DNA sequences:

Raw sequence data is available at NCBI SRA: mock community, SRX4908948, the Malaise sample, SRX5175597, and the gradient PCR experiment, SRX4908947.

## Data Availability

Scripts are available as Supplemental Files, and the metabarcoding pipeline is available at GitHub https://github.com/VascoElbrecht/JAMP.

PrimerMiner sequence alignments are available on Dryad (10.5061/dryad.249rk92).

## Supplemental Information

Supplemental information for this article can be found online at http://dx.doi.org/10.7717/peerj.7745#supplemental-information.

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
