# Peer review of "Validation of COI metabarcoding primers for terrestrial arthropods"

_PeerJ, doi:10.7717/peerj.7745_

## Round 0.1 · original submission · Minor Revisions

Dear Vasco and Colleagues,

I have received three independant assessments of your manuscript. All reviewers have recognised the quality and value of this study and have offered a number of minor revisions - well done!

I agree with reviewer 2 that adding a more direct statement about which primer sets are recommended and for which applications would be valuable - if you agree.

I will be looking forward to receiving your revised manuscript along with a point-by-point response to the reviewers comments.

With warm regards,
Xavier

·

Basic reporting

The manuscript is well written and clear to understand. There is suffcient literature cited.

Experimental design

The research question was well defined and performed to a high standard.

One question was if the authors sequenced negatives and if so how these were dealt with in the bioinformatics.

Validity of the findings

The findings were valid and the conclusions reached backed up by the results presented.

Additional comments

L62: While this is true COI databases are also limited in taxonomic groups outside of insects.

Figure 1: Braukman 2018 or 2019?

L113: Mock community extracts were ….

L127: (( not needed

L172: Were sequencing negatives run?

L192: Figure 3 overviews the results of the processing steps……

L229: Amplicon concentrations (ng/ul) reached …..

Reviewer 2 ·

Basic reporting

no comment

Experimental design

no comment

Validity of the findings

no comment

Additional comments

I have reviewed the manuscript entitled “Validation of COI metabarcoding primers for terrestrial arthropods” by Elbrecht and colleagues. In this work the authors evaluated a large number of potential primer sets for the amplification and identification of bulk arthropod samples using DNA metabarcoding. Their systematic approach to evaluation involved gradient PCR to determine amplification efficiency at differing TA, sequencing to determine detection of full species richness, and annealing temperature optimization for a subset of primer sets. They go on to recommend several sets of primers that performed well and advocate for the use of single primer sets for arthropod identification as opposed to the use of multiple primer sets to amplify all species.

Overall, I found this work to be impressive in its thoroughness, both in terms of the study design and the level of supporting documentation provided. This is a valuable piece of work for the molecular ecology community and I commend the authors for its undertaking. My comments are mostly minor with one major suggestion.

Major comment:

The authors have gone to great lengths to evaluate a large number of primer sets for arthropod identification, but the concluding statements do not seem definitive. “Our study demonstrates that the fwh2, BF1/2/3 + BR2 and mlCOIintF based primer sets all perform well when metabarcoding terrestrial arthropod samples. For most of these primer sets, annealing temperatures of 46-50°C are ideal. When data is analyzed for Exact Sequence Variants(ESVs), the BF3 + BR2 primer set is recommended as it is not affected by primer slippage.”

I think the impact of this paper would be improved with more direct statements about which primer sets are recommended and for which applications. An abbreviated version of these statements should appear in abstract and in the conclusions. This paper is likely to be cited specifically for these recommendations and they should be easily accessible to the reader.

For example: “We recommend fwhF2 + fwhR2n for amplification of degraded DNA samples such as gut contents, and BF3 + BR2 when high taxonomic resolution is required”.


Minor comments:

Line 52 – Change “on” to “of”

Line 55 – change “targets” to “target”

Line 79 – Check journal style, but there is a missing “)” or citation brackets should be removed.

Line 84 – Suggest adding… given the increased cost and time “associated with amplifying and sequencing additional markers”…

Line 113 – Change “These” to “The”

Line 122 - Insert brief description of the utility of gradient PCR for this study. How is this tool being used to identify best primer sets? Describe that the heat block itself provides a gradient of temps so a full plate provides a range of TA across replicate reactions.

Line 129 – No replication? A statement here about why replication was not performed would be useful. I know this is addressed in the discussion, but you can also briefly justify the decision here.

Line 143 – Please clarify what the exact selection criteria were.

Line 149 – I found the structure of this section confusing. I suggest clearly breaking this into paragraphs for each topic, e.g., fusion primer design, PCR 1, PCR 2, library prep, sequencing.

Line 150 – change “where” to “were”

Line 154 – What is meant by “sequenced half the samples reverse orientation”? Meaning the primer tags were reversed, or adapter sequences? Please clarify.

Line 173 – Why was less DNA used for the Malaise trap sample?

Line 184 – Please translate code functions. What is fastq_pctid?

Line 206 – Again, please specify the exact selection criteria that were used.

Line 218 – The subsampling/bootstrapping process in Figures 4 and 5 is not described in this section. Please explain.

Line 263 – Fig S11 rarefaction curves. This may be worth moving to the main MS as it is a good visual representation of the main findings. I understand there is some conceptual overlap with Fig 4 but the shape of these curves clearly conveys the advantages of certain primer sets.

Line 319 – This statement seems overly bold. Sample replication is clearly very important, but PCR/Sequencing replication can also be important for generating consensus taxonomic assignment at the sample level and for identifying contamination.

Line 331 – Does this reflect sequencing bias or DNA degradation? The same pattern has been observed in DNA quantity by fragment length using qPCR of degraded samples.

Lines 352 – 354 – Is it also possible that primers for shorter fragments pick up more rare species because trace DNA is more amplifiable at short lengths? This could be an argument for the use of fwhF2 + fwhR2n or short barcoding regions with high taxonomic resolution.

Line 371 – Change “predicted” to predict

Line 423 – Change “,” to “;”

Line 440 – It is not clear from my reading of this MS why one would choose a primer set with a longer fragment over fwhF2 + fwhR2n. Short fragments have many advantages and if this provides similar taxonomic resolution to the longer fragments then why choose a longer fragment?

Lines 441 – Change “fwh2” to “fwhF2”

Line 451 – I commend the authors for providing extensive supplemental information. However, in my quick scan of the R code I did not find it very well annotated. If the objective is simply to document what was done this may be sufficient, but if the goal is for others to use this code then addition annotation would be useful.

·

Basic reporting

The manuscript is written in excellent English. The structure is standard for a scientific article. The literature background is enough and all cited references are relevant.

Experimental design

The research question is well introduced. The experimental design is impeccable. The investigation is performed with the highest technical standards, rigorous scientific interpretation of the results, and transparency.

Validity of the findings

The recommendations derived from this paper will significantly improve the area in the future. The interpretation of the results are rigorous and absent of any speculation.

Additional comments

This manuscript by Elbrecht et al. is a long-expected, exhaustive and impeccably designed comparison of many primer sets previously proposed for metabarcoding of terrestrial arthropod communities. This information is invaluable to clarify the current chaos with a huge variety of primer sets for arthropod metabarcoding, which were proposed by different authors, many of them without any previous critical evaluation. The conclusions and recommendations presented in this paper will significantly improve the consistency of metabarcoding projects of arthropod communities in the future and will facilitate the comparison of results obtained from different workflows in the past. The authors are clearly committed to transparency in their results, so they made an effort to provide an exhaustive set of supplementary information files, which can be very helpful for many researchers working in metabarcoding primer design and evaluation. I strongly recommend the publication of this work in PeerJ. I include some minor comments and recommendations (mainly to correct some typos) below.

Minor comments:

L34: Correct “demonstrated” to “demonstrates”, to keep congruence with the present tense forms.

L 80-87: An important additional caveat of using multiple primer sets could be discussed here. Namely, the difficulty to merge read abundances obtained using multiple primer sets in order to extract some quantitative value, which basically makes multi-primer metabarcoding techniques a presence/absence detector, without any interpretable quantitative value.

L127: Remove double parentheses in ((Vamos et al. 2017)

L132: What particular materials were used as positive and negative controls? This should be explained further here.

L150: Correct “where selected” to “were selected”

L176: Add the “n” to “an Illumina MiSeq”

L229-230: Correct “Amplicon concentrations reached (ng/ul)” to “Amplicon concentrations (ng/µl) reached”

L233: Correct “Amplification success for newly designed primers was mixed” to “Amplification success for newly designed primers showed mixed results”

L270: “The primer combinations fwhF2 + fwhR2n, BF2/BF3 + BR2, ArF5 + Fol-degen-rev and mlCOIintF”. The mlCOIintF primer (and its derivatives) is used in actually 3 “good” combinations. Also BF1 (which has a good BF1+BR2 combination and is targetting almost the same “Leray fragment”) is missing from the sentence. So I would rephrase this sentence to: “The primer combinations fwhF2 + fwhR2n, BF1/BF2/BF3 + BR2, ArF5 + Fol-degen-rev and those based on mlCOIintF and its derivatives”.

L271: Correct “recovery =< 95%” to “recovery ≥ 95%”.

L349: “Although these longer fragments improve taxonomic resolution”. This improvement is probably not significant in practice at the species level, since the analysed COI fragment is practically saturated after 200 bp. I propose to change the sentence to “Although these longer fragments might improve taxonomic resolution”.

L371: Change “These complexities are difficult to predicted” to “These complexities are difficult to predict”.

L372 and L388: “in vivo”. You mean “in vitro”. An “in vivo” metabarcoding procedure would imply all the flies, bees and bettles flying around the lab, while we sequence them.

L375. A recent publication by Collins et al. (2019) is very relevant to illustrate this non-specific amplification issue and could be added to the reference list:
Collins, R.A., Bakker, J., Wangensteen, O.S., Soto, A.Z., Corrigan, L., Sims, D.W., Genner, M.J. and Mariani, S., 2019. Non‐specific amplification compromises environmental DNA metabarcoding with COI. Methods in Ecology and Evolution, in press. Doi: 10.1111/2041-210X.13276

L392: Correct “PCR is also biased” to “PCR can be also biased”

L409: Correct “metabarcoding primer” to “metabarcoding primers”.

L409: Correct “given that of most” to “given that most”

L447-448. “As a general rule, the use of multiple primer sets seems rarely justified as it increases laboratory effort without substantially improving taxon recovery” I suggest to add: “As a general rule, the use of multiple primer sets seems rarely justified as it increases laboratory effort and adds complexity to the interpretation of read abundance, without substantially improving taxon recovery”

L458: Don’t forget to add the DOI to access the PrimerMiner results in Dryad.

Figure 1 (caption):
Delete “the” in “four primer sets showing the good performance”

Figure 1: Correct “Asses” to “Assess”. Correct “tempratures” to “temperatures” (2 times).

Figure 4: I think that the red cross symbol used to indicate “primer slippage” in this figure can be misleading to the readers, who could erroneously think that these primer sets are not recommendable to be used at all. I suggest changing this red cross to a symbol with less negative connotations, such as “*” or “PS”, and using a different, more neutral, color for this symbol.
I think that primer slippage is not really so big an issue as the authors are implying in this figure. Primer sets affected with primer slippage can be readily used in practice and the recovered sequences differing in just a leading base (or in 2 leading bases) are never a big problem, since they can be easily dealt with by most (if not all) taxonomy assignment algorithms, specially those involving a final taxonomic clustering step. Even if an ESV approach is used, the right ESVs can be recovered and assigned to the right taxon in the reference database, so that an easy and quick final step of taxonomic clustering would fix this minor issue and would produce a right taxon list.

---

## Round 0.2 · accepted · Accept

Dear Vasco and co-authors,

I am pleased to accept your revised manuscript for publication in PeerJ. I have attached an annotated version of the ms with suggested modifications to be incorporated in the proofs, as you see fit.

This is a very comprehensive and impressive work which will represents a hugely valuable contribution to the field - Thank you!

With warm regards,
Xavier

Reviewer 2 ·

Basic reporting

no comment

Experimental design

no comment

Validity of the findings

no comment

Additional comments

The authors have done an excellent job of integrating the feedback from all reviewers. I have no additional comments or concerns. I look forward to seeing the paper in print.

·

Basic reporting

Nothing to add.

Experimental design

Nothing to add.

Validity of the findings

Nothing to add.

Additional comments

I am happy with the changes introduced by the authors. Well done and congratulations.